# Three-Dimensional Model Improves Body Image Perception After Bariatric Surgery

**DOI:** 10.3390/jcm14134787

**Published:** 2025-07-07

**Authors:** Cyril Gauthier, Matthieu Poussier, Célia Lloret-Linares, Marc Danan, Anamaria Nedelcu

**Affiliations:** 1EMNO Bourgogne, ESPD Nuvee, 4 rue Lounes Matoub, 21000 Dijon, France; 2EMNO Drôme-Ardèche, Clinique Kenedy, 26200 Montélimar, France; m.poussier@emno.fr; 3Hôpital Privé Pays de Savoie, 74100 Annemasse, France; celia.lioret-linares@ramsaysante.fr; 4Clinique Saint Michel, 83100 Toulon, France; mdanan@wanadoo.fr (M.D.); anamaria.andreica@gmail.com (A.N.)

**Keywords:** bariatric surgery, three-dimensional imaging, body satisfaction

## Abstract

**Background**: Despite losing weight, the majority of subjects retained an obese view of themselves. The aim of the study was to evaluate the usefulness of a 3D modeling tool in improving the body image of patients who have undergone bariatric surgery. **Methods**: Morbidly obese subjects involved in a medico-surgical obesity management program and having undergone a Roux en Y Gastric Bypass (RYGB) or a sleeve gastrectomy (SG) were prospectively included during their usual postoperative medical follow-up. The figure rating scale (FRS), body image questionnaire, and Hospital Anxiety Depression Scale test were performed. The FRS was assessed before and after visualizing their body image using a 3D modeling tool. Distributions between the groups for gender (female vs. male) and type of surgery (gastric bypass vs. sleeve gastrectomy) were tested with a Pearson’s chi^2^ independence test. The significance threshold was *p* < 0.05. **Results**: We included 140 adults with sleeve gastrectomy (72.9%; *n* = 102) or gastric bypass (27.1%; *n* = 38). The mean time from surgery was 308.3 ± 111.4 days (63–511). Participants were mostly female (77.9%; *n* = 109). Nearly half of the subjects who had undergone bariatric surgery almost one year before modified their body perception after visualizing their avatar thanks to a 3D modeling tool. One third reduced their FRS score (“perceived body”) after visualizing their avatar. FRS score and body mass index (BMI) following surgery (“real body”) were significantly correlated before and after visualizing the 3D avatar, with a stronger correlation after visualizing the 3D avatar. **Conclusions**: A 3D modeling tool may improve body perception after weight loss in subjects with bariatric surgery. Being simple, non-invasive, not expansive, and easy to use during a consultation and to understand for the patient, a regular use of this tool may be largely implemented in clinical practice. Its usefulness in improving body image, mood disorders, and eating disorders and the further success of the surgery should be further evaluated.

## 1. Introduction

Since 1980, obesity rates have more than doubled worldwide. By 2014, over 600 million adults—approximately 13% of the global adult population—were classified as obese. Bariatric surgery has emerged as an important treatment for severe obesity, offering significantly better results in weight loss, comorbidity reduction, and mortality rates compared to non-surgical interventions, regardless of the procedure type [1,2]. According to a 2014 survey by the International Federation for the Surgery of Obesity and Metabolic Disorders (IFSO), 579,517 bariatric surgeries were performed worldwide that year. Sleeve gastrectomy (SG) was the most commonly performed procedure, followed by Roux-en-Y gastric bypass (RYGB) and adjustable gastric banding (AGB) [3]. On average, one-year and five-year total weight loss percentages were 31.2% and 25.5% for RYGB, 25.2% and 18.8% for SG, and 13.7% and 11.7% for AGB [4].

Morbid obesity is strongly associated with profound psychosocial consequences, including depression, anxiety, and diminished self-esteem [5]. These mental health challenges are often compounded by body image dissatisfaction, which is influenced by a combination of psychological, behavioral, and perceptual factors that can deteriorate over time. Even after significant weight loss through bariatric surgery, many individuals continue to perceive themselves as obese, struggling to reconcile their new physical appearance with their self-image. This persistent misperception can hinder the psychological adjustment necessary for long-term surgical success [6].

The difficulty in adapting to changes in body image and identity is closely linked to dissatisfaction and an impaired quality of life. Patients who experience this struggle often face ongoing psychological distress, which can undermine their motivation, compromise adherence to lifestyle changes, and ultimately impact the long-term effectiveness of the surgery [7]. Addressing these psychological challenges through counseling, support groups, and cognitive-behavioral therapy is crucial to help patients achieve a healthier self-image and sustain the benefits of bariatric surgery. A holistic approach that integrates psychological support with surgical intervention is key to improving patient satisfaction, quality of life, and long-term outcomes.

Body image is a complex, multidimensional concept encompassing both the evaluation of one’s physical appearance and the perception of body shape and size. This aspect plays a critical role in the holistic approach to managing patients who undergo bariatric surgery. Positive body image can greatly influence psychological well-being, self-esteem, and overall satisfaction with surgical outcomes. However, body image concerns are frequently overlooked in post-bariatric care, even though significant changes in physical appearance often follow these procedures.

While various therapeutic tools and intervention protocols have been developed to address altered body image, their application has been limited in the context of bariatric surgery patients [8,9,10]. These tools typically aim to help individuals reconcile their perception of themselves with their new, post-surgical reality. In this context, the use of advanced 3D modeling technology has emerged as a promising method for enhancing body image satisfaction. By providing patients with a realistic visualization of their evolving body shape, 3D modeling can help bridge the gap between perception and reality, facilitating a more accurate self-image and promoting psychological adaptation.

The objective of this study was to evaluate the potential benefits of integrating a 3D modeling tool into the care of patients who have undergone bariatric surgery. This study aims to investigate the impact of a 3D body imaging tool on the perception of body image in post-bariatric surgery patients, with a focus on identifying perceptual discrepancies and psychological patterns associated with body image changes.

## 2. Materials and Methods

### 2.1. Study Population

Morbidly obese subjects involved in an obesity management program (Clinique du Chalonnais; Hôpital Privé Sainte Marie; Hôpital Privé Dijon Bourgogne; Clinique Convert; Centre Hospitalier William Morey, France) and having undergone a RYGB or an SG were prospectively included in the current study. In the usual postoperative period, subjects benefit from educational programs during which the difficulties of perceiving bodily changes are addressed (Figure 1). Subjects who were pregnant, had visual impairment, and were not able to maintain a vertical position were not evaluated.

The current study utilizes a cross-sectional observational design to assess body image perception among post-bariatric surgery patients using 3D imaging. The protocol was approved by the institutional review board at French Ramsay Group Research Foundation, and all participants provided informed consent.

### 2.2. Body Weight Measurements

Data for body mass and body mass index obtained during the follow-up outpatient visit (BM-FUP and BMI-FUP, respectively), and at the time of the bariatric surgery (BM-SURG and BMI-SURG) were collected. These data were also used to calculate (1) the percentage of body mass index loss (pBMI) between the surgery and the follow-up sessions, and (2) the rate of body mass index loss (“Rate-pBMI”), based on the time between the date of surgery and that of the outpatient follow-up visit.

### 2.3. Assessment of Anxiety and Depression

Patients underwent the Hospital Anxiety Depression Scale test [11]. This test uses 14 items scaled from 0 to 3 to identify symptoms of anxiety or depression, with 7 items each. A total score of ≥11 represents a potential disorder.

### 2.4. Assessment of Body Image

The Stunkard figures scale [12], also called the figure rating scale (FRS score), was used. This scale is a visible measure of how an individual perceives his or her own physical appearance. Each figure presents nine male and nine female schematic silhouettes, ranging from extreme thinness (score = 1) to extreme obesity (score = 9). Participants are asked to self-select the silhouette that best indicates his or her current body size.

The body image questionnaire developed by Koleck et al. was used [13]. This allows the evaluation of desirable and undesirable body attributes or states, based on 19 items. Responses are made according to a scale of scores between 1 and 5, where 1 and 5 correspond to directly opposed terms (e.g., physically attractive body vs. physically unattractive body). The sum of the responses gives a score between 19 and 95 (the QIC score), where higher scores reflect a more positive body image.

In addition, patients answered the following questions:-Prior to visualizing the avatar (“Q1”): “Are you aware of your physical change?”-After visualizing the avatar (“Q2”): “Did the avatar allow you to become aware of your physical change?”

Responses were based on a scale ranging from 0 to 6, where 0 corresponds to “not at all”, and 6 corresponds to “completely”.

### 2.5. Computerized 3D Modeling of Body Silhouettes

NETTELO 3DMG is a Windows PC software (Windows 10) allowing scanning and analyzing anyone in 3 dimensions from 2 photos (front and profile) (Figure 2). The person in underwear stands at about 2 m from the camera with their arms slightly away from the body. Nettelo’s 3D reconstruction technology allows us to obtain measurements with a precision of 1 cm.

Three groups were determined according to the difference between the FRS score before and after visualizing the 3D avatar (also called FRS-pre and FRS-post, respectively):Group A: FRS-pre = FRS-post, respectively;Group B: FRS-pre > FRS-post, respectively;Group C: FRS-pre < FRS-post, respectively.

### 2.6. Statistical Analyses

Qualitative ordinal variables (FRS-pre, FRS-post, Q1, Q2, QIC, HAD) were analyzed using a Kruskal–Wallis non-parametric test to identify differences between groups A, B, and C. After verification of normal distribution with a Kolmogorov–Smirnov test, quantitative variables (BMI-FUP, BMI-SURG, BMI loss, Rate-pBMI, age, and time between the surgery and the date of the follow-up outpatient visit) were evaluated with a triple-factor non-parametric (Kruskal–Wallis) or parametric (ANOVA) test using (1) group (A vs. B vs. C), (2) gender (female vs. male), and (3) type of surgery (sleeve gastrectomy versus gastric bypass). When a significant effect was seen, a post hoc Fisher LSD test was performed. Linear regression line analyses were performed for the variables BMI-FUP and FRS-pre, as well as for BMI-FUP and FRS-post, and Spearman’s rank correlation coefficients were determined. Distributions between the groups A, B, and C for gender (female vs. male) and type of surgery (gastric bypass vs. sleeve gastrectomy) were tested with a Pearson’s chi^2^ independence test. The significance threshold was *p* < 0.05. Mean ± standard deviation (SD) is reported in the text, and median ± standard error (SE) in the figures.

## 3. Results

### 3.1. Population Characteristics—Demographic Data

We included 140 adults with sleeve gastrectomy (72.9%; *n* = 102) or gastric bypass (27.1%; *n* = 38). The mean time from surgery was 308.3 ± 111.4 days (63–511). Participants were mostly female (77.9%; *n* = 109).

### 3.2. Body Image Perception Outcomes

Nearly half of the patients (48%; 67 of 140) modified their FRS score (perceived body) after visualizing their avatar. Among these 67 patients, 49 (35%; group B) had a reduced score and 18 (13%; group C) had an increased score. The three groups of patients were similar in terms of sex ratio, age, types of surgeries, delay after surgery, anthropometric data before and after surgery, body image questionnaire, depression and anxiety, and QIC score. As expected and as shown in Figure 3, the FRS-pre values for groups A, B, and C (5.00 ± 0.93, 6.22 ± 0.81, and 4.06 ± 0.75, respectively) were significantly different (*p* < 0.001), while those for FRS-post were not.

FRS score (“perceived body”) and BMI-FUP (“real body”) were significantly correlated before and after visualizing the 3D avatar, with a stronger correlation after visualizing the 3D avatar (Figure 4).

When adjusted on BMI, FRS was different between groups before seeing the avatar but similar after.

### 3.3. Three-Dimensional Analysis Findings

#### Self-Evaluation Questions on the Perception of Physical Change and Usefulness of the 3D Tool

The self-evaluation question relating to patients’ perception of physical change (Q1: “Are you aware of your physical change?”) showed that the patients in group B expressed greater difficulty in perceiving physical change, compared to patients in group A (*p* <0.01). Concerning the usefulness of the 3D avatar (Q2), 95% of the cohort attributed the maximum score of 6.

## 4. Discussion

Our current data support the potential benefits of integrating a 3D modeling tool into the care of patients included in a bariatric program. This additional tool will help the patient to enhance body image perception, improve patient satisfaction, and support long-term psychological adjustment and well-being. Well-being with mood disorders is frequently discussed in the bariatric literature, and weight loss does not result in consistent long-term reduction in their prevalence. Some studies have observed a higher number of cases of mood disorders and suicides after surgery, in comparison with the general population [14,15]. Factors other than the degree of weight loss contribute to the risk of depression and altered quality of life. It is important to identify and take care of them as mood disorders may impact long-term outcomes of bariatric surgery [16]. In our cohort, approximately 40% of patients had a HAD score over 11, suggesting a significant number of patients with mood disorders after surgery.

Our results showed that nearly half of the subjects who underwent bariatric surgery experienced an improvement in body perception after visualizing their avatar through a 3D modeling tool. Additionally, one-third of participants reported a reduction in their FRS score (“perceived body”) following this visualization. Notably, FRS scores and BMI (“real body”) were significantly correlated both before and after viewing the 3D avatar, with a stronger correlation observed post-visualization.

Among the postoperative risk factors of mood disorders associated with bariatric surgery, results of surgery may not match with patient’s expectations. Improved satisfaction with body shape is frequently reported in patients having undergone weight loss surgery [17,18]. But some subjects may have unrealistic expectations, while others may overestimate their corpulence. Hrabosky et al. [18] reported that changes in weight and body image relate poorly to each other, suggesting that mediating factors may be involved. As for individuals suffering from obesity, this may be a precursor to psycho-pathologic issues and notably eating disorders, which are associated with poorer weight loss after surgery [19]. As one third of the patients in the study had a reduced FRS score after visualizing their avatar, we can imagine that the new perception of body corpulence may improve body satisfaction and the overall satisfaction with the bariatric surgery and reduce eating disorders. Indeed, patients recognize the usefulness of the tool. It would be interesting to evaluate the effect of more frequent use of this tool during weight loss on body satisfaction. Interestingly, a lower proportion of subjects with a HAD score over 11 are represented in group C, where subjects have a leaner perception of their corpulence. These may be patients with better psychological health before and/or after surgery or subjects feeling a high benefit of surgery independently of the degree of body weight loss, such as resolution of hypertension, diabetes or obstructive apnea syndrome, and improvement of musculoskeletal pain.

Several therapeutic tools and protocols have been developed to improve the management of altered body image: the identification of one’s silhouette, perception of body space, or distorted images [8]. Use of computerized 3-dimensional (3D) modeling technologies of human silhouettes have been used to help individuals suffering from eating disorders and altered body image [20]. Unlike the classic photographic images from which patients tend to form a subjective judgment their body, the construction of an avatar may allow a more detached analysis, which is more favorable for objective appreciation of the corpulence.

In addition, various programs may contribute to improved body satisfaction. Exercise interventions have shown effectiveness in enhancing psychological well-being. Similarly, online programs, such as three body functionality-focused writing exercises, have demonstrated a positive impact on women’s body image by emphasizing body functionality [21,22]. This approach to fostering a positive body image may also be applicable to bariatric subjects.

It is important to consider the context of the current study. All 140 patients included in our study followed a therapeutic educational program in which difficulties in perception of body image after surgery were presented, anticipated, and addressed. This may partly explain the relative coherence between the perception of the silhouette using the Stunkard figures scale and the avatar image for more than half (group A). There may be a higher number of patients with difficulties in body change perception in the populations of subjects having undergone bariatric surgery, and the use of an avatar may benefit a large number of subjects whenever it is used during the postoperative period. Unfortunately, we did not evaluate the HAD and FRS scores before surgery, and we cannot identify in this study the subgroup of patients that may largely benefit from the use of the avatar.

The results of this ongoing prospective study have allowed us to make progress on the dynamic assessment of the correlation between the patient’s actual morphology and their perceived body image (discrepancy), as well as its impact on psychological and social factors and the overall effectiveness of obesity treatment. Based on the conclusions of this study, LIPOLINE, a company specializing in custom perioperative abdominal compression, has developed an innovative approach within the standard obesity management pathway through a specific patient support program. This approach, which is both neuropsychological and morphological, utilizes 3D avatar generation tools, online HAD tests, and quality of life assessments (at home) in both preoperative and postoperative phases. These resources are accessible via an HDS-compliant platform for anonymized data processing and retrieval, and they are the objective of a multicentric prospective clinical trial.

Our current manuscript has several limitations that warrant discussion. First, one of the major limitations is the absence of a control group. As this is a retrospective study, the inclusion of a control group would be more appropriate in a prospective design. Additionally, the follow-up period is limited; given the nature of bariatric surgery and its long-term implications, extended follow-up is essential to fully understand the evolution of body image perception over time. Finally, the potential impact of excess skin following significant weight loss (an important factor that can influence body image) was not assessed longitudinally, which may limit the comprehensiveness of our findings.

## 5. Conclusions

In conclusion, it seems beneficial to use a 3D modeling tool to improve body perception after weight loss in the bariatric population. Being simple, non-invasive, not expansive, and easy to use during a consultation and easy to understand for the patient, regular use of this tool may be largely implemented in clinical practice. Its usefulness in improving body image, mood disorders, and eating disorders and the further success of the surgery should be further evaluated.

## Figures and Tables

**Figure 1 jcm-14-04787-f001:**
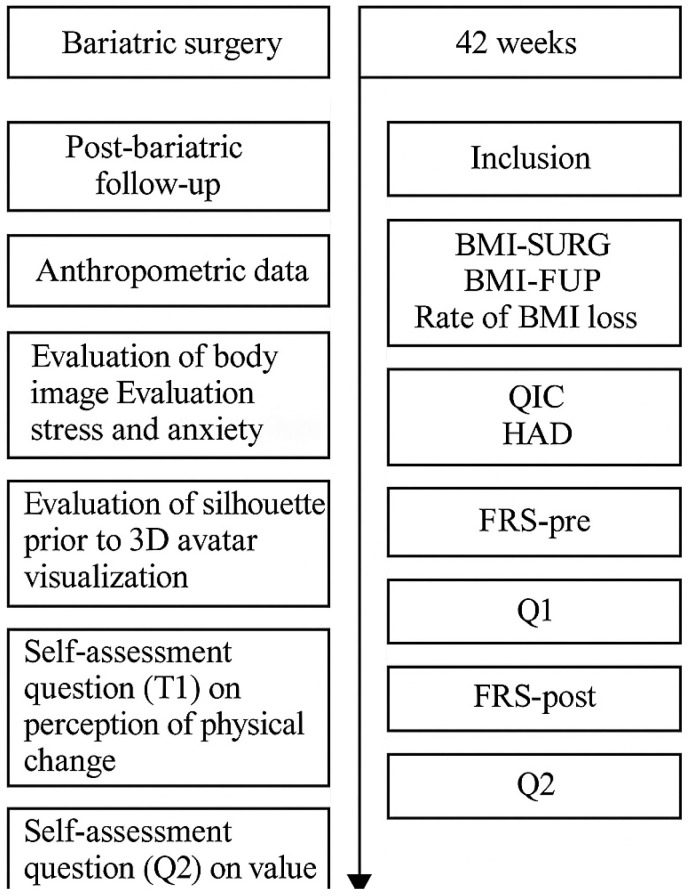
BMI-FUP: outpatient follow-up body mass index; BMI-surg: body mass index before bariatric surgery; pBMI: percentage of BMI loss; Tx-BMI: percentage of BMI loss per week; QIC: body image questionnaire; HAD: Hospital Anxiety and Depression Scale; FRS-pre: Stunkard figure rating scale before 3D avatar visualization; FRS-post: Stunkard figure rating scale after 3D avatar visualization.

**Figure 2 jcm-14-04787-f002:**
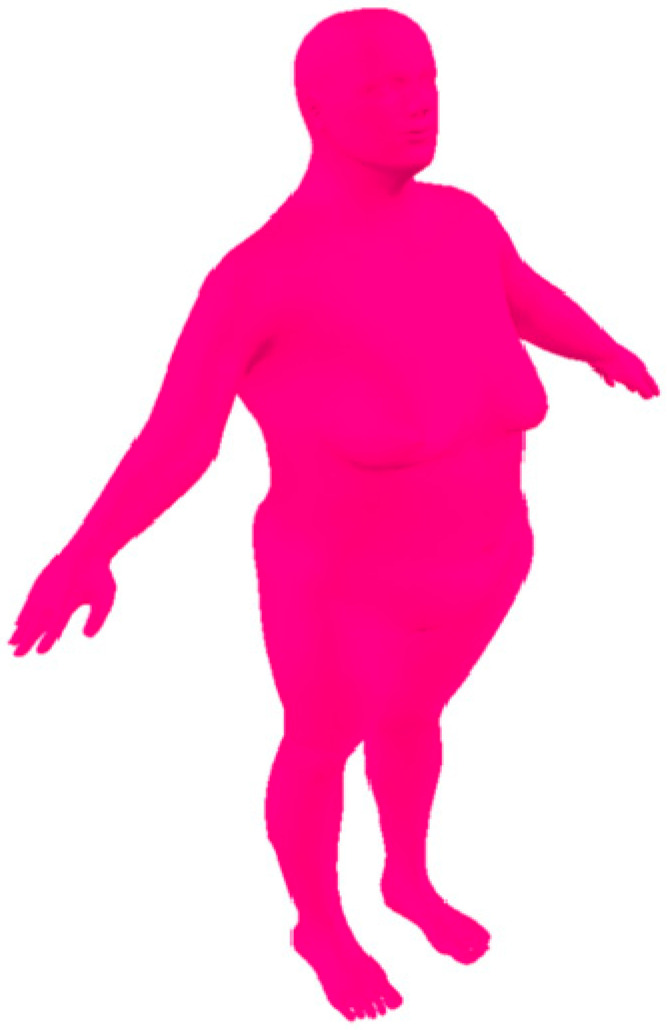
Example of a computerized 3D modeling of the body silhouette of a patient participating in the study.

**Figure 3 jcm-14-04787-f003:**
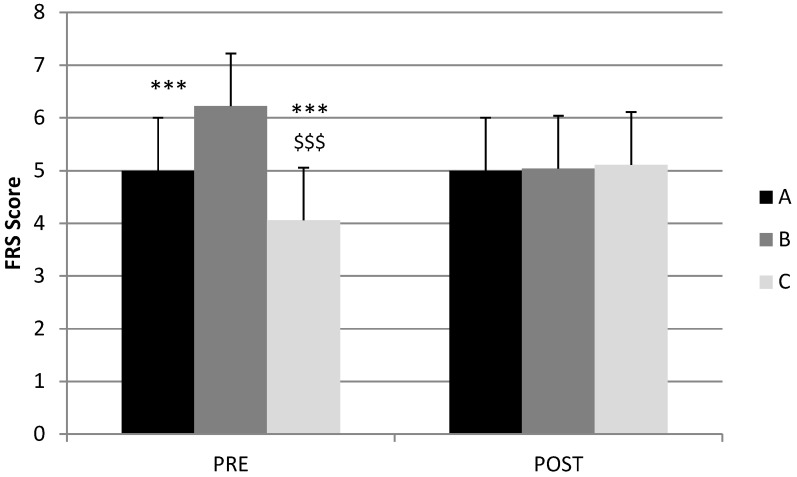
Stunkard scale scores recorded before (FRS-pre) and after (FRS-post) 3D avatar visualization for patient groups A, B, and C (Mean ± SE). *******: significantly different from group B (*p* < 0.001); $$$: significantly different from group A (*p* < 0.001). Group A: FRS-pre = FRS-post. Group B: FRS-pre > FRS-post. Group C: FRS-pre < FRS-post.

**Figure 4 jcm-14-04787-f004:**
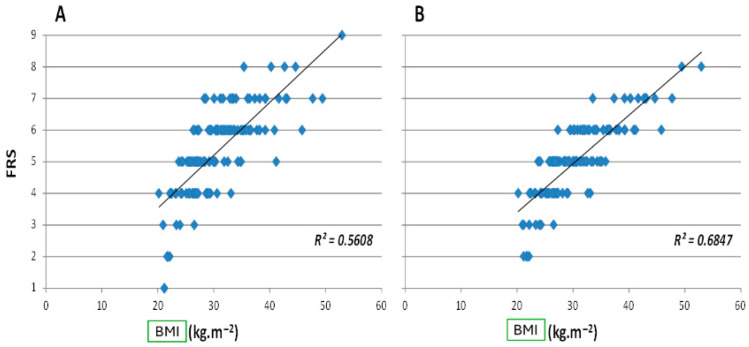
Linear regressions established for all subjects between the FRS score (i.e., “perceived body”) and BMI-FUP (i.e., “actual body”) before 3D avatar visualization ((**A**); FRS-pre) and after 3D avatar visualization ((**B**); FRS-post).

## Data Availability

The original contributions presented in this study are included in the article. Further inquiries can be directed to the corresponding author(s).

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
