# Peer review of "Three-Dimensional Model Improves Body Image Perception After Bariatric Surgery"

_jcm, 2025, doi:10.3390/jcm14134787_

Round 1

Reviewer 1 Report

Comments and Suggestions for Authors

Abstract

The aim is better to be declared before the methodology.

It is advised to spell out abbreviations upon their first appearance in the manuscript.

Brief description of statistical testing method should be declared in the abstract.

Please add supporting figure of your findings ( such as differences in proportions, P values ) to indicate the magnitude of the change and its statistical significance in the abstract.

Introduction

The authors made the following statement:

‘Even after significant weight loss through bariatric surgery, many individuals continue to perceive themselves as obese, struggling to reconcile their new physical appearance with their self-image. This persistent misperception can hinder the psychological adjustment necessary for long-term surgical success.’

Is this statement supported by an evidence or is it an observation introduced by the authors?

The authors made the following statement:

‘This approach aims to enhance body image perception, improve patient satisfaction, and support long-term psychological adjustment and well-being.’

This statement is not suitable as a research aim or objectives as it seems more associated with clinical / intervention recommendations

Please consider revision

Materials and methods

What is the research design of the study? Please elaborate.

Figure one: please modify the language of the figure to English

Sample size estimation was not provided. Please explain what was the sample size required to answer the study objectives.

It is currently difficult to evaluate suitability of declared statistical tests because the design of the study was not declared. Is this an interventional study?

Results

Writing of the results is currently confusing. The authors need to clearly display the main subheading of different results section.

The authors are strongly encouraged to add tables to the results section. The current narration of their findings is very confusing and incoherent .

Titles of figure two, three, and four should be changed to English

Figures three and four are not well displayed and some sections are missing. Please  revise.

Discussion/ conclusions

It is advised to start the discussion with a very brief summary of the study aim, and the main findings. This is a common practice as recommended by some reporting guidelines.

It is odd to start the discussion with mood disorders and suicide although it was not investigated in the current study.

The limited clarity concerning study design, display of the results made it difficult to assess the suitability of the discussion.

Comments on the Quality of English Language

English language editing service is needed to enhance the writing quality of the manuscript.

Author Response

We thank the reviewers for the positive feedback. We are convinced that by the modifications done to the manuscript according to your previous suggestions we have highly improved the quality of our paper.

Reviewer # 1

Abstract

The aim is better to be declared before the methodology.

Thank you for this suggestion. We have now clearly stated the study aim at the beginning of the abstract.

It is advised to spell out abbreviations upon their first appearance in the manuscript.

All abbreviations have now been spelled out at first mention in the abstract and throughout the manuscript.

Brief description of statistical testing method should be declared in the abstract.

A short description of the statistical methods used has been added to the abstract.

Introduction

The authors made the following statement:

‘Even after significant weight loss through bariatric surgery, many individuals continue to perceive themselves as obese, struggling to reconcile their new physical appearance with their self-image. This persistent misperception can hinder the psychological adjustment necessary for long-term surgical success.’

Is this statement supported by an evidence or is it an observation introduced by the authors?

 This statement is supported by Reference 6, which discusses the persistence of distorted body image and self-perception issues in post-bariatric surgery patients. We have clarified this in the manuscript by directly linking the statement to the cited evidence and ensuring the reference is appropriately contextualized.

The authors made the following statement:

‘This approach aims to enhance body image perception, improve patient satisfaction, and support long-term psychological adjustment and well-being.’

This statement is not suitable as a research aim or objectives as it seems more associated with clinical / intervention recommendations

Please consider revision

We completely agree with you and please find in the revised form of the manuscript the new form of the phrase to clearly distinguish between the research aims and broader clinical implications. The revised objective now focuses on investigating changes in body image perception using the 3D tool.

"This study aims to investigate the impact of a 3D body imaging tool on the perception of body image in post-bariatric surgery patients, with a focus on identifying perceptual discrepancies and psychological patterns associated with body image changes." 

Materials and methods

What is the research design of the study? Please elaborate.

We have clarified in the revised manuscript that this study utilizes a cross-sectional observational design to assess body image perception among post-bariatric surgery patients using 3D imaging.

Figure one: please modify the language of the figure to English

Figure 1 has been updated, and all labels and captions have been translated into English for clarity and consistency.

Sample size estimation was not provided. Please explain what was the sample size required to answer the study objectives.

All this data can be retrieved in the results section.

It is currently difficult to evaluate suitability of declared statistical tests because the design of the study was not declared. Is this an interventional study?

 We confirm that this is not an interventional study. The design is observational and exploratory. This has been clearly stated in the revised methods section.

Results

Writing of the results is currently confusing. The authors need to clearly display the main subheading of different results section.

We have restructured the results section by adding subheadings to clearly distinguish different areas of analysis (e.g., demographic data, body image perception outcomes, 3D analysis findings).

Titles of figure two, three, and four should be changed to English

We apologies for this error. All figure titles and captions have been translated into English

Figures three and four are not well displayed and some sections are missing. Please  revise.

Figures 3 and 4 have been reformatted for improved clarity and presentation. We ensured all components are fully visible

Discussion/ conclusions

It is advised to start the discussion with a very brief summary of the study aim, and the main findings. This is a common practice as recommended by some reporting guidelines.

It is odd to start the discussion with mood disorders and suicide although it was not investigated in the current study.

The discussion now opens with a concise summary of the study objectives and key findings, in line with best practices and reporting guidelines. You can find in the revised form of the manuscript the revised paragraph: “Our current data supports the potential benefits of integrating a 3D modeling tool into the care of patients included in a bariatric program. This additional tool will help the patient to enhance body image perception, improve patient satisfaction, and support long-term psychological adjustment and well-being”

The limited clarity concerning study design, display of the results made it difficult to assess the suitability of the discussion.

We have clarified the study design and restructured the results as noted above, which in turn has allowed us to refine and streamline the discussion section accordingly

Reviewer 2 Report

Comments and Suggestions for Authors

The article is well written, and the use of a 3-D representation of the subjects is a nice novelty.
The sample size is good, but it would have been interesting to have more male subjects so that comparisons of results between the sexes could also be made.
The major limitations to the study are a lack of a control group and absent follow-up.

- I recommend that all figures be corrected as they are barely legible (Figure 1) or completely on the edge of the paper.
- The captions are all in French so I could not understand them, sorry.
- I did not understand if the following were also considered:
1) The presence of excess skin after weight loss, which could affect the perception of body image;
2) Assessing whether improved body image perception results in positive behavioral changes or better clinical outcomes.

Author Response

We thank the reviewers for the positive feedback. We are convinced that by the modifications done to the manuscript according to your previous suggestions we have highly improved the quality of our paper.

Reviewer # 2

The article is well written, and the use of a 3-D representation of the subjects is a nice novelty.

We appreciate this positive feedback and have retained the 3D visual elements, improving figure clarity as suggested.

The sample size is good, but it would have been interesting to have more male subjects so that comparisons of results between the sexes could also be made.

We thank you very much for your comment, we acknowledge this limitation, but the current population could not be modified. Future research will aim to include a more balanced gender distribution to allow for sex-specific analysis.

The major limitations to the study are a lack of a control group and absent follow-up.

We thank you again for your comment and we have suggested directions for future longitudinal and comparative studies.

- I recommend that all figures be corrected as they are barely legible (Figure 1) or completely on the edge of the paper.

All figures have been reformatted to improve resolution, size, and placement within the manuscript.

- The captions are all in French so I could not understand them, sorry.

We apologies for this inconvenient and we have corrected our error in the initial form of the manuscript. We have translated all captions and in-figure text to English for accessibility to a broader audience.

- I did not understand if the following were also considered:

1) The presence of excess skin after weight loss, which could affect the perception of body image;
2) Assessing whether improved body image perception results in positive behavioral changes or better clinical outcomes.

These important considerations were not part of the current study’s scope, but we now mention them in the limitations and future directions section, acknowledging their potential impact on body image perception and the need for further research.

The following paragraph was included in the revised form of the manuscript:

“Our current manuscript has several limitations that warrant discussion. First, one of the major limitations is the absence of a control group. As this is a retrospective study, the inclusion of a control group would be more appropriate in a prospective design. Additionally, the follow-up period is limited; given the nature of bariatric surgery and its long-term implications, extended follow-up is essential to fully understand the evolution of body image perception over time. Finally, the potential impact of excess skin following significant weight loss (an important factor that can influence body image) was not assessed longitudinally, which may limit the comprehensiveness of our findings.”

Round 2

Reviewer 1 Report

Comments and Suggestions for Authors

I thank the authors for their effort in addressing all comments.